# Zinc Oxide Nanoparticles Improve Salt Tolerance in Rice Seedlings by Improving Physiological and Biochemical Indices

Abhishek Singh [1,*], Rakesh Singh Sengar [1,*], Vishnu D. Rajput [2], Tatiana Minkina [2] and Rupesh Kumar Singh [3,4]

1. Department of Agricultural Biotechnology, College of Agriculture, Sardar Vallabhbhai Patel University of Agriculture and Technology, Meerut 250110, Uttar Pradesh, India
2. Academy of Biology and Biotechnology, Southern Federal University, 344090 Rostov-on-Don, Russia; rajput.vishnu@gmail.com (V.D.R.); tminkina@mail.ru (T.M.)
3. Department of Protection of Specific Crops, InnovPlantProtect Collaborative Laboratory, Estrada de Gil Vaz, Apartado 72, 7350-999 Elvas, Portugal; rupeshbio702@gmail.com
4. Centre of Molecular and Environmental Biology, Department of Biology, Campus of Gualtar, University of Minho, 4710-057 Braga, Portugal
* Correspondence: intmsc.abhi@gmail.com (A.S.); sengarbiotech7@gmail.com (R.S.S.); Tel.: +91-880-095-56-71 (A.S.); +91-941-247-22-92 (R.S.S.)

**Abstract:** Understanding the salinity stress mechanisms is essential for crop improvement and sustainable agriculture. Salinity is prepotent abiotic stress compared with other abiotic stresses that decrease crop growth and development, reducing crop production and creating food security-related threats. Therefore, the input of metal oxide nanoparticles (NPs) such as zinc oxide nanoparticles (ZnO-NPs) can improve salt tolerance in crop plants, especially in the early stage of growth. Therefore, the aim of the current study was to evaluate the impact of ZnO-NPs on inducing salt tolerance in two rice (*Oryza sativa* L.) genotypes of seedlings. An undocumented rice landrace (Kargi) and salinity tolerance basmati rice (CSR 30) seeds were grown in a hydroponic system for two weeks with and without 50 mg/L concentrations of ZnO-NPs in various doses of NaCl (0, 60, 80, and 100 mM). Both Kargi (15.95–42.49%) and CSR 30 (15.34–33.12%) genotypes showed a reduction in plant height and photosynthetic pigments (chlorophyll a and b, carotenoids, and total chlorophyll), Zn content, and $K^+$ uptake under stress condition, compared with control seedlings. On the other hand, stress upregulated proline, malondialdehyde (MDA), $Na^+$ content, and antioxidant enzyme activities—namely, those of superoxide dismutase (SOD), ascorbate peroxidase (APX), catalase (CAT), and glutathione reductase (GR)—in both *O. sativa* genotypes over the control. However, ZnO-NP-treated genotypes (Kargi and CSR 30) restored the photosynthetic pigment accumulation and $K^+$ level, reforming the stomata and trichome morphology, and also increased antioxidant enzymes SOD, APX, CAT, and GR activity, which alleviated the oxidative stress, while reducing the level of MDA, proline, and $H_2O_2$ under stress condition. The present findings suggest that adding ZnO-NPs could mitigate the salinity stress in *O. sativa* by upregulating the antioxidative system and enhancing the cultivation of undocumented landrace (Kargi) and basmati (CSR 30) genotypes of *O. sativa* in salinity-affected areas.

**Keywords:** salinity; malondialdehyde; superoxide dismutase; ascorbate peroxidase; catalase; glutathione reductase; Kargi; CSR 30; zinc oxide nanoparticles; abiotic stress

## 1. Introduction

The human population is rapidly increasing, and it may reach approximately 10 billion by 2050. This increases in population may enhance food demand [1,2]. Continuously growing human population and changes in climate due to global warming may affect the prevailing farming practices, reducing crop production [2,3]. Currently, one-third of world agricultural lands suffer from salinization due to inappropriate agrochemical and

agriculture practices, and industrial pollution also exacerbates this problem [4,5]. Salinity is causing an increase in devastating stress conditions due to climate change; approximately 6% of fertile lands, directly and indirectly, turn into uncultivable [6,7]. It is a catastrophic stressor that affects plant growth by disturbing its physio-biochemical processes, nutrient uptake, and ionic homeostasis by developing osmotic pressure and ion toxicity [6,7]. Soil with a saline character severely hampers plant growth and its processes of development [8].

Usually, salinity stress induces initial osmotic stress due to the excessive accumulation of ions. However, damage at the subcellular level is also influenced by extreme reactive oxygen species (ROS) generation, including superoxide radical ($O_2^{\bullet-}$), singlet oxygen ($^1O_2$), hydrogen peroxide ($H_2O_2$), and hydroxyl radical ($OH^\bullet$) accumulation at the higher concentration in plant tissues due to hyperosmotic stresses and ionic imbalance [7,9]. Excessive generation of ROS leads to lipid oxidation and produces malondialdehyde (MDA), negatively impacting cellular metabolism and physiological activities. This resulting adverse effects on membrane integrity, which affect biological functions of membranes such as maintaining cellular integrity and transport of molecules, cell signaling, and compartmentalization of cellular processes [10]. Besides hyperosmotic stress and ionic disturbance, salinity also creates a nutritional imbalance that causes a reduction in photosynthetic activities by affecting the photosynthetic pigments [11]. At the physiological level, salinity stress affects the leaf water potential. It creates a water deficit in plants, leading to stomatal closure and a decreased gas exchange process with a higher risk of oxidative injury [12]. Stomatal closure decrease in gas exchange can reduce photosynthesis [12].

Plant types also define the level of salinity tolerance, and, for example, halophytes are more adaptive under saline conditions, while glycophytes are sensitive [13,14]. Salinity tolerance in glycophytic crops, including rice (*Oryza sativa*), is mainly associated with maintaining ion homeostasis by lowering the $Na^+/K^+$ or high $K^+/Na^+$ ratios by excluding $Na^+$ ions from the plant cell [15]. In addition to maintaining ion homeostasis, glycophytic plants develop various strategies such as an increase in the accumulation of compatible solutes such as proline, which plays a vital role in adaptation under osmotic stress through stabilizing the tertiary structure of proteins [16–18]. Proline is a nontoxic osmoprotectant with a low molecular weight that acts as a signaling molecule, helps in stabilizing the cellular structure, and scavenges the ROS produced during salinity stress [19]. During salinity stress, various antioxidants alleviate salinity's effect on the plant species' survival [20]. Inherent antioxidant enzymes in plants—namely, superoxide dismutase (SOD), ascorbate peroxidase (APX), catalase (CAT), and glutathione reductase (GR)—also play crucial roles in preventing the toxicity effects generated by ROS generation.

An increase in oxidative stress reduces the antioxidant enzyme activity and enhances the MDA and $H_2O_2$ levels [21–23]. This complex relationship between antioxidant enzyme activity and the detoxification process for salinity tolerance has been studied in some plant species, including *O. sativa* [24–27]. Sahi et al. [28] demonstrated that *O. sativa* is more sensitive to salinity stress in an early stage of growth than in the reproductive stage. Thus, the initial phase of *O. sativa* growth is sensitive to salinity, which affects its physiological, morphological, and biochemical indices. The salinity stress at the seedling stage consequently affects the overall growth and yield of *O. sativa* [29,30]. More than 3.5 billion people worldwide rely on *O. sativa* for nutrient intake, and salinity stress has become a major threat to food security [31–33]. Therefore, in the present context, there is a need for new methods or emerging approaches to protect the plants, especially *O. sativa*, at the seedling stage for better quality yield to meet the increasing food demand [34,35].

Nanotechnology has the potential to maintain sustainable agriculture practices [36–38], and metal-based nanoparticles (NPs), especially zinc oxide nanoparticles (ZnO-NPs), showed promising results for crop growth and have been used in various formulations and forms such as nanofertilizers, seed priming, or foliar inputs [37,39]. Zinc is an important micronutrient for plants [2]. Moreover, Zn is also an essential co-factor in different biocatalytic enzymes, e.g., hydrolases, ligases, isomerases, oxidoreductases, and transferases [38], and have been tested at a physio-biochemical level in various model plants

including *O. sativa* [40–49]. Currently, ZnO-NPs is vitally used in multiple commercial practices that may benefit crop production and the protection of agriculture-related process [50]. The application of ZnO-NPs can alleviate the harsh environmental stress effects on plants. Under salinity stress, treating ZnO-NPs can improve the plant's morphological, physiological, and biochemical characteristics [2,51–53]. Moreover, limited studies have documented the impact of ZnO-NPs on *O. sativa* and rarely on the early stage of its growth under salinity stress [54–59]. Therefore, considering the above facts, as well as the need for insights into physio-biochemical activities influenced by NPs in stress conditions, the present study aimed to investigate the responses of ZnO-NPs on undocumented, rare, and on-the-verge-of-extinction *O. sativa* landrace (Kargi) and salt-tolerant variety of basmati rice (CSR 30) under different doses of salts at an early stage of seeding growth in controlled conditions, i.e., using a hydroponic system, for precise results.

## 2. Results

### 2.1. Effect of ZnO-NPs on the Early Stage of Rice under Salinity Stress Condition

Under salinity stress, a significant reduction in plant height was observed, by 15.95–42.49% for Kargi and 15.34–33.12% for CSR 30, compared with nonstressed *O. sativa* seedlings (Table 1). By input of ZnO-NPs, a very slight decrease in the height was observed in Kargi (14.62–28.1%) and CSR 30 (12.01–27.10%) (Table 1). However, the reduction rate in *O. sativa* seedling height in the Kargi genotype was slightly higher than that in CSR 30 in terms of both ZnO-NPs and stress conditions. At the same time, improvement was observed in 100 mM NaCl with ZnO-NPs, compared with the corresponding treatment without ZnO-NPs. However, the other two salinity treatments, i.e., 60 and 80 mM NaCl, did not show any significant difference, compared with their complementary ZnO-NPs treatment.

**Table 1.** Effects of ZnO-NPs on *Oryza sativa* seedling height (cm) of Kargi and CSR 30 genotypes under salinity stress conditions.

| *Oryza sativa* Genotype | Control | 60 mM NaCl | 80 mM NaCl | 100 mM NaCl | 60 mM NaCl + ZnO-NPs (50 mg/L) | 80 mM NaCl + ZnO-NPs (50 mg/L) | 100 mM NaCl + ZnO-NPs (50 mg/L) |
|---|---|---|---|---|---|---|---|
| Kargi | 15.07 ± 0.12 [a] | 12.67 ± 0.45 [b] | 10.43 ± 0.37 [c] | 8.67 ± 0.13 [d] | 12.87 ± 0.36 [b] | 11.03 ± 0.27 [c] | 10.83 ± 0.09 [c] |
| CSR 30 | 15.27 ± 0.55 [a] | 12.73 ± 0.46 [b] | 11.67 ± 0.17 [c] | 9.6 ± 0.27 [d] | 13.43 ± 0.33 [b] | 11.53 ± 0.09 [c] | 11.13 ± 0.40 [c] |

Values are the means of three replicates' standard deviation (±SD). Different small letters (a–d) within the row indicate a statistically significant value (LSD) at $p \leq 0.01$ among the treatments.

### 2.2. $Na^+$ and $K^+$ Concentrations Analysis

An increased $Na^+$ and decreased $K^+$ amount were observed, compared with control, under salinity stress.

Applying ZnO-NPs alleviated the salinity stress by decreasing the $Na^+$ and increasing the $K^+$ uptake. Under salinity stress conditions, the $Na^+$ concentration significantly increased between 337.74% and 754.72% in Kargi, and 247.27% and 608.73% in CSR 30 leaves, compared with control (Figure 1A). After the application of ZnO-NPs, a reduction in $Na^+$ concentration was found in the leaves of Kargi (166.04–418.87%) and CSR 30 (120–321.82%) in comparison to salt-stressed and nonstressed control *O. sativa* seedling (Figure 1A).

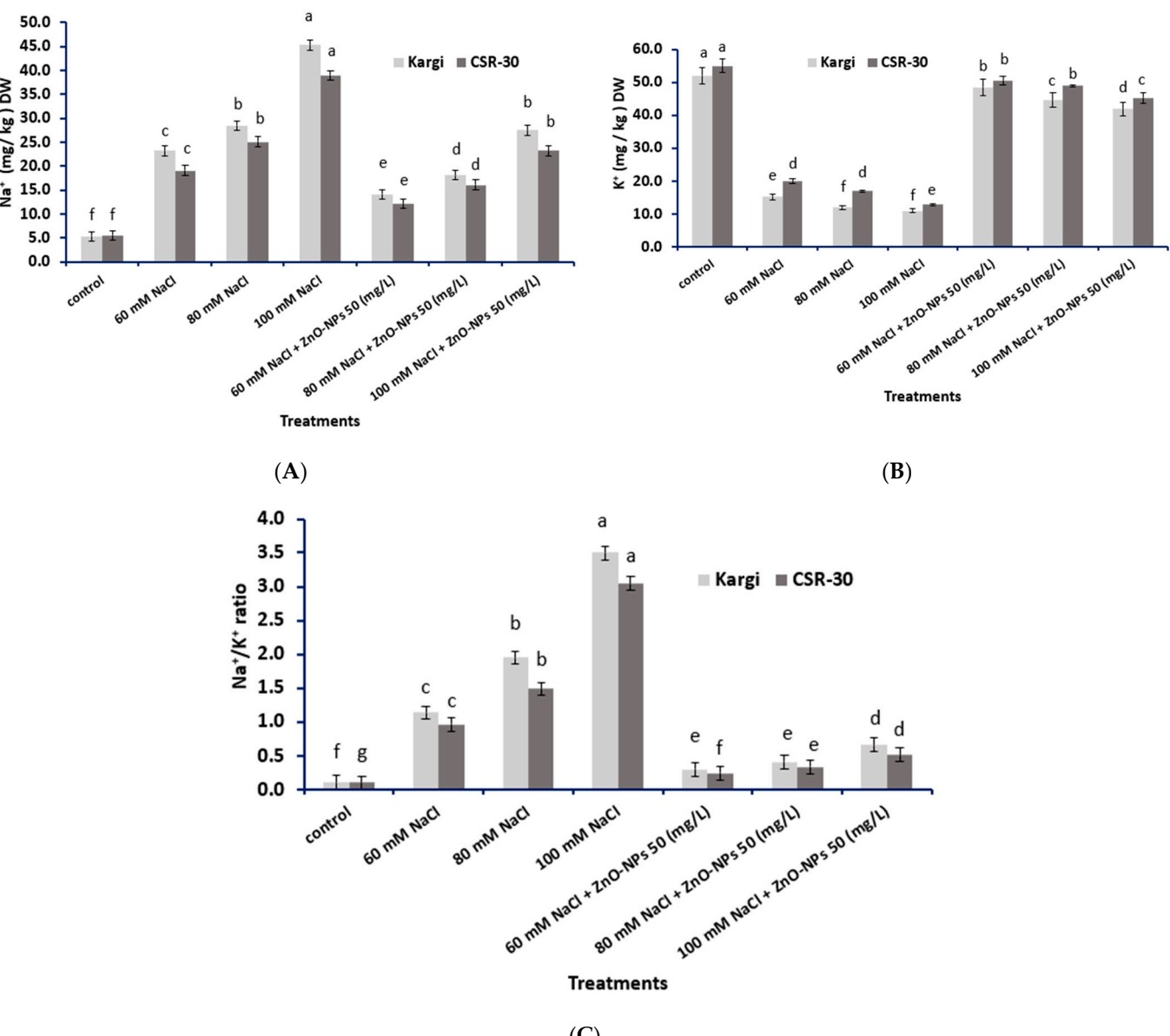

**Figure 1.** Impact of ZnO-NPs on (**A**) Na$^+$ (**B**) K$^+$ and (**C**) Na$^+$/K$^+$ ratio concentration in leaves of Kargi and CSR 30 *Oryza sativa* genotype in control and saline conditions. Different small alphabets (a–g) indicate a statistically significant difference between treatments concerning control. Values are means of three replicates. Error bars indicate the least considerable value (LSD) at $p \leq 0.01$ among the treatments.

Under salinity stress, the level of K$^+$ decreased in leaves of Kargi (70.96–79.04%) and CSR 30 (63.82–76.73%), respectively, in comparison to control (Figure 1B). However, after the input of ZnO-NPs, the K$^+$ concentration was 8.73–19.42% in Kargi and 6.18–17.64% in CSR 30 leaves, compared with salt-stressed and nonstressed plants, respectively (Figure 1B).

The observation of the Na$^+$/K$^+$ ratio indicated a higher Na$^+$/K$^+$ ratio in Kargi and CSR 30 under the saline conditions, and minimum changes were observed in ZnO-NP-treated *O. sativa* seedling (Figure 1C). However, in NaCl and ZnO-NP treatments, a slight increase in Na$^+$/K$^+$ ratio was observed in Kargi, compared with CSR 30 genotypes of *O. sativa* (Figure 1C).

### 2.3. Proline Accumulation

The increased proline accumulation in the leaves of Kargi and CSR 30 was noted in salinity stress conditions. It rose from 240.91% to 363.64% in Kargi and from 225% to 375.02% in CSR 30 under stress conditions, compared with control seedlings (Figure 2). Under saline conditions with the combination of ZnO-NPs, the level of proline in leaves decreased in Kargi from 36.36% to 104.55% and in CSR 30 from 25% to 100%, compared with the control treatment (Figure 2). Less proline accumulation in ZnO-NP-treated Kargi and CSR 30 leaves indicates physiological adaptation under saline conditions. A high accumulation of proline in Kargi, compared with that in CSR 30, was found with both saline and ZnO-NP treatments.

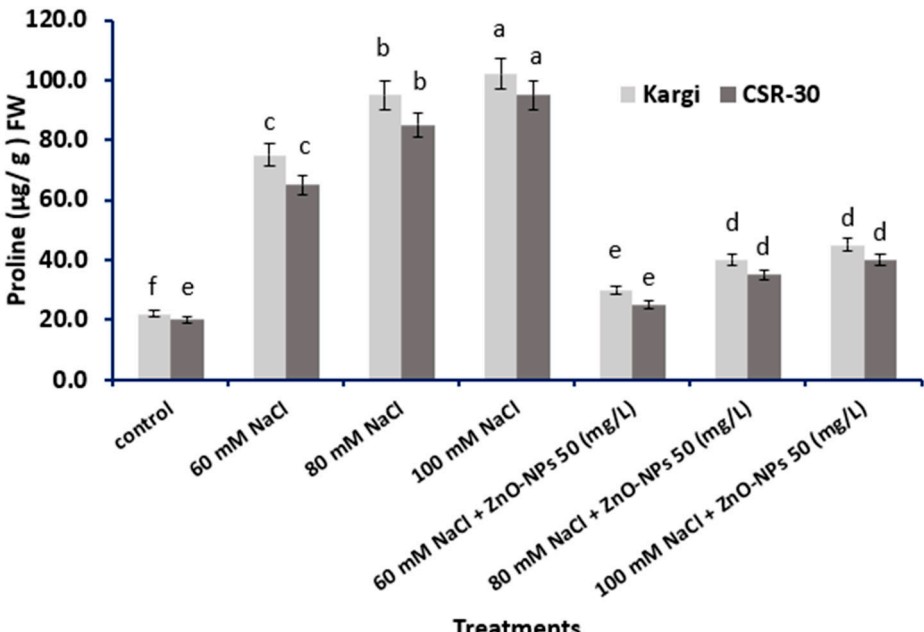

**Figure 2.** Impact of ZnO-NPs on proline accumulation in Kargi and CSR 30 leaves in control and under saline conditions. Values are means of three replicates; a statistically significant difference between treatments compared with control is indicated by different small alphabets (a–f). Error bars indicate the least considerable value (LSD) at $p \leq 0.01$ among the treatments.

### 2.4. Hydrogen Peroxide ($H_2O_2$) Content

High accumulation of $H_2O_2$ was observed under saline conditions, which was manifested as 50–150% in Kargi and 50–133.33% in CSR 30, respectively, compared with control treatments (Figure 3). However, a significant decline in $H_2O_2$ accumulation was found in ZnO-NP treatments between 16.67% and 50% in Kargi and between 20% and 50% in CSR 30 leaves, compared with nonstressed control *O. sativa* seedlings (Figure 3).

### 2.5. MDA (Malondialdehyde) Content

The MDA level was increased in leaves of Kargi (17.20–47.31%) and CSR 30 (15.23–45.23%) under salinity stress (Figure 4). However, the addition of ZnO-NPs varied the MDA content significantly between 9.67% and 19.35% in Kargi and between 8.85% and 12.50% in CSR 30 under salinity stress, compared with control plants (Figure 4). Specifically, the MDA concentration in Kargi was slightly higher than that in CSR 30 with both saline and ZnO-NP treatments. Conversely, in ZnO-NP treatments, the concentration of MDA was decreased in both *Oryza sativa* genotypes.

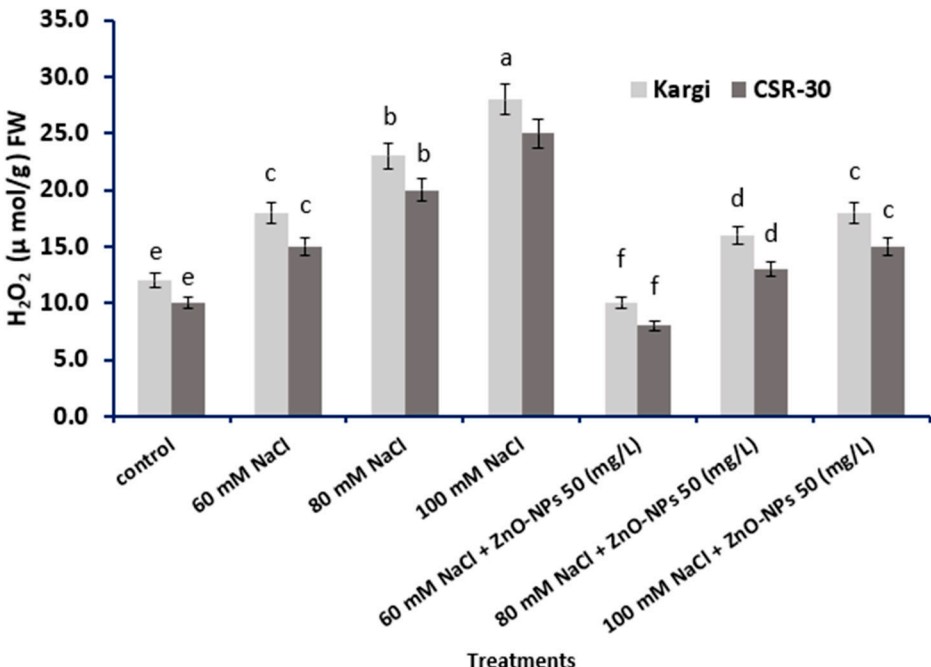

**Figure 3.** Effect of ZnO-NPs on $H_2O_2$ content in leaves of Kargi and CSR 30 Oryza sativa genotype in control and under saline conditions. Values are means of three replicates. Different small alphabets (a–f) indicate a statistically significant difference between treatments compared with control. Error bars indicate the least considerable value (LSD) at $p \leq 0.01$ among the treatments.

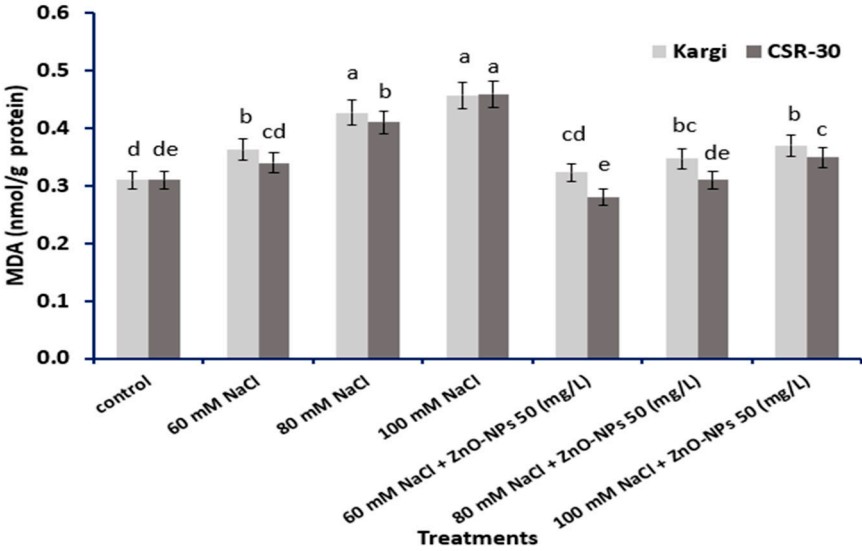

**Figure 4.** Impact of ZnO-NPs on MDA content in leaves of Kargi and CSR 30 Oryza sativa genotype in control and under saline conditions. Values are means of three replicates; a statistically significant difference between treatments compared with control is indicated by different small alphabets (a–e). Error bars indicate the least considerable value (LSD) at $p \leq 0.01$ among the treatments.

### 2.6. Zn Content

In saline treatments, accumulation of Zn content was significantly decreased, by 18.20–56.52% in Kargi and 12.82–48.72% in CSR 30 leaves, compared with nonstressed plants (Figure 5). After the addition of external Zn supply in the form of ZnO-NPs, increased Zn content was observed in Kargi (212.82–133.08%) and CSR 30 (212.50–139.13%) leaves (Figure 5). Zn accumulation in saline and ZnO-NP treatments was slightly lower in Kargi than in CSR 30.

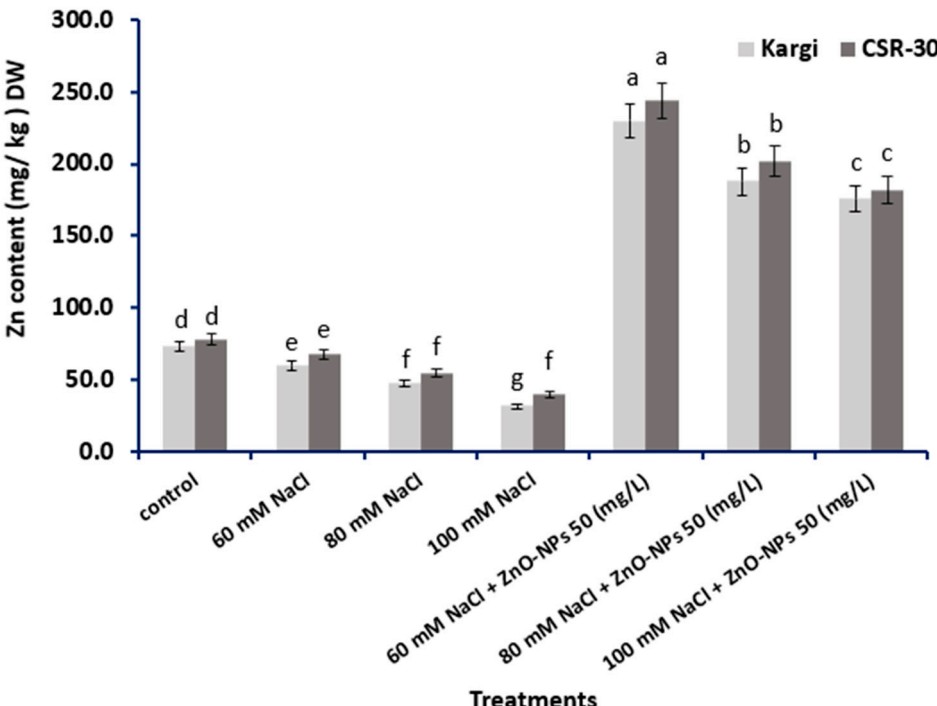

**Figure 5.** Impact of ZnO-NPs on Zn content in leaves of Kargi and CSR 30 Oryza sativa genotype in control and under saline conditions. Small alphabets (a–g) indicate a statistically significant difference between treatments concerning control. Values are means of three replicates. Error bars indicate the least significant value (LSD) at $p \leq 0.01$ among the treatments.

*2.7. Photosynthetic Pigments Content*

A noticeable reduction was noted in photosynthetic pigments (Chl a, Chl b, total chlorophyll, and carotenoids) under stress conditions, compared with nonstressed treatments. However, the application of ZnO-NPs increased the level of these photosynthetic pigments under saline conditions.

Under salinity stress, photosynthetic pigments significantly decreased in Kargi (Chl a by 58.23–36.71%, Chl b by 61.52–69.21%, total chlorophyll by 51.92–54.48%, and carotenoids by 35.54–54.55%) and in CSR 30 (Chl a by 45.20–26.12%, Chl b by 42.13–32.01%, total chlorophyll 53.11–53.18%, and carotenoids 35.86–32.63%), compared with nonstressed treatments (Figure 6). However, when treated with ZnO-NPs, photosynthetic pigments were enhanced (Figure 6).

*2.8. Antioxidant Enzyme Activity*

A clear change was found in the antioxidant enzyme activity in Kargi and CSR 30 with saline and ZnO-NP treatments. NaCl-treated Kargi and CSR 30 seedlings showed increased antioxidant enzyme activity, compared with those using nonstressed treatments. Furthermore, using ZnO-NPs enhanced antioxidant enzyme activities, compared with both saline and nonsaline treatments.

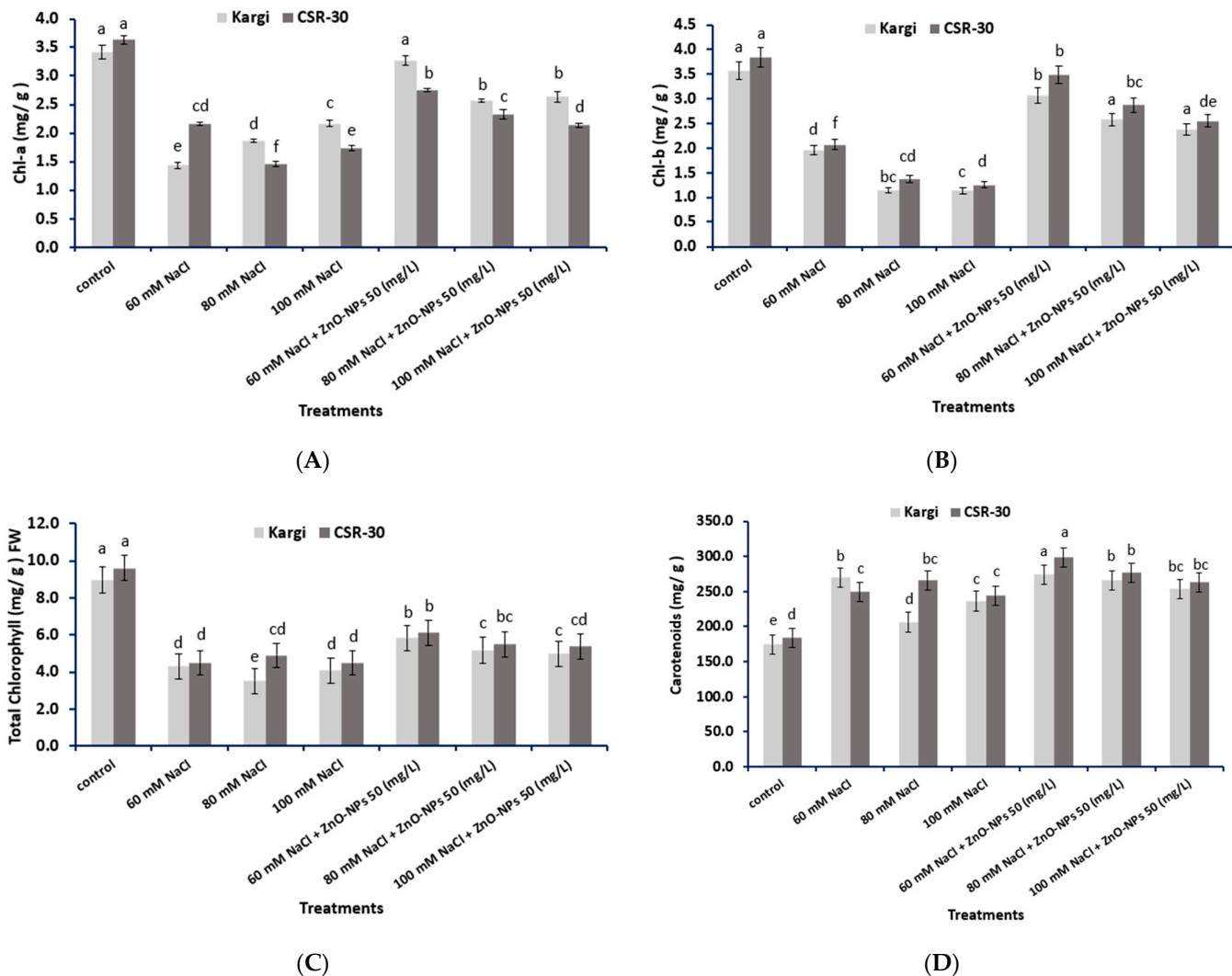

**Figure 6.** Impact of ZnO-NPs on photosynthetic pigments (**A**) Chl a, (**B**) Chl b, (**C**) total chlorophyll, and (**D**) carotenoids in leaves of Kargi and CSR 30 *Oryza sativa* genotypes in control and under saline conditions. Different small alphabets (a–f) indicate a statistically significant difference between treatments compared with control. Values are means of three replicates. Error bars indicate the least significant value (LSD) at $p \leq 0.01$ among the treatments.

### 2.8.1. SOD (EC 1.15.1.1)

The SOD functions as the first line of defense under stress conditions, which significantly increased under saline conditions, by 23.69–31% in leaves of Kargi and 35.51–52.06% in CSR 30, compared with that observed with nonstressed treatment (Figure 7A). Treatment with ZnO-NPs showed a further increment in SOD. The activity of SOD in leaves was noted by 37.58–60.99% in Kargi and 47.42–70.23% in CSR 30, respectively, compared with that found with stressed and nonstressed treatments (Figure 7A). Compared with all treatments, the activity of SOD was slightly less in Kargi than in CSR 30.

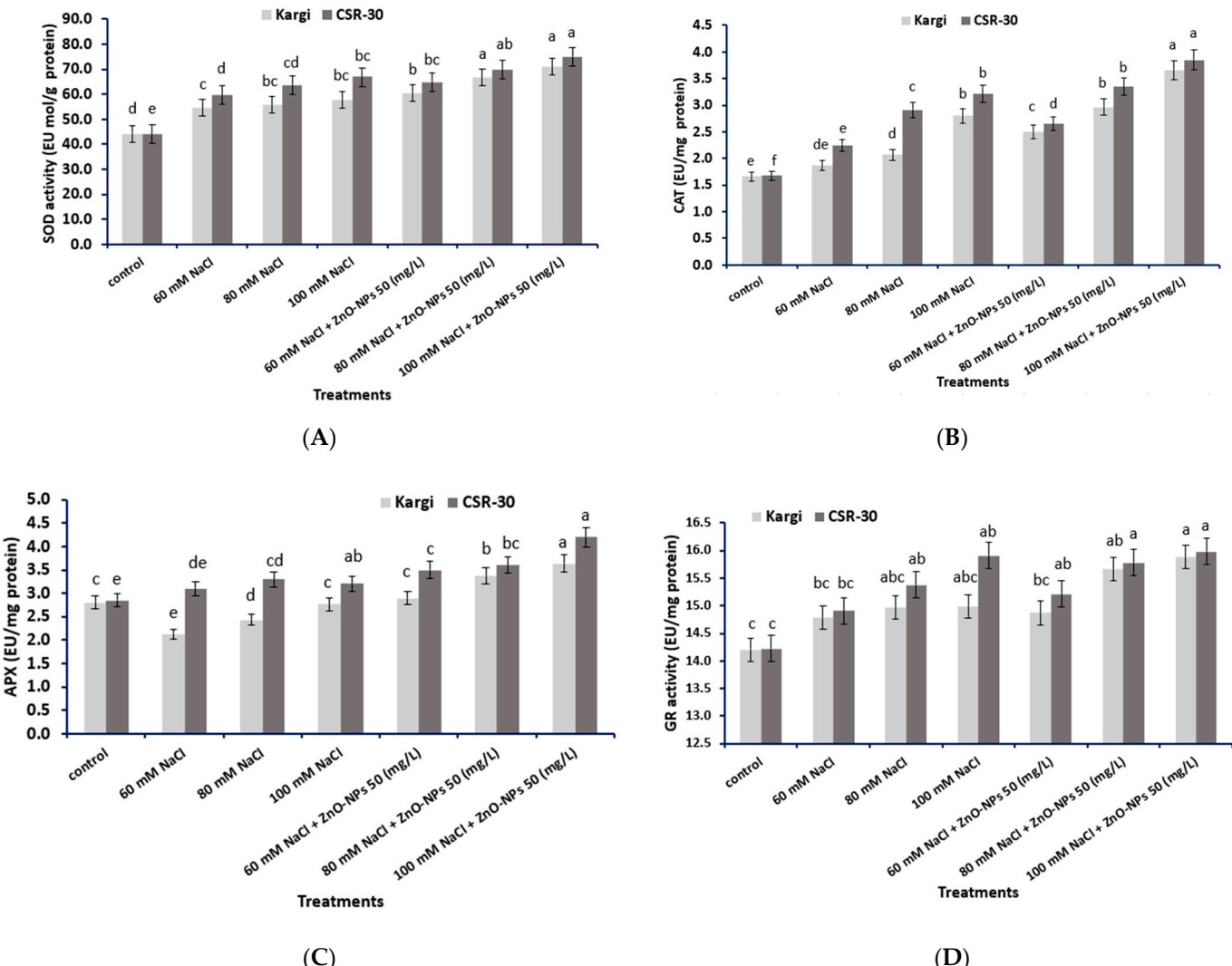

**Figure 7.** Impact of ZnO-NPs on antioxidant enzymes (**A**) SOD, (**B**) CAT, (**C**) APX, and (**D**) GR activity in leaves of Kargi and CSR 30 *Oryza sativa* genotypes in control and under saline conditions. Different small alphabets (a–f) indicate a statistically significant difference between treatments compared with control. Values are means of three replicates. Error bars indicate the least significant value (LSD) at $p \leq 0.01$ among the treatments.

### 2.8.2. CAT (EC 1.11.1.6)

Under saline conditions, a marked increase in CAT activity was found in the leaves of Kargi (12.65–68.67%) and CSR 30 (33.93–91.07%) in stress conditions, compared with that found with nonstressed treatments (Figure 7B). All treatments showed a noticeable increase in CAT activity under stressed and nonstressed environments. However, a prominent increase was revealed in ZnO-NP treatments of Kargi (50.40–120.48%) and CSR 30 (57.74–129.17%), compared with their saline and untreated control seedlings (Figure 7B). CAT activity was slightly lesser in all the treatments of Kargi than that of CSR 30.

### 2.8.3. APX (EC 1.11.1.11)

In all of the saline treatments, a significantly high level of APX activity was shown in Kargi (13.21–32.45%) and CSR 30 (15.79–36.84%) leaves, compared with that found with nonsaline treatment (Figure 7C). All saline treatments showed a remarkable increase in APX activity in both genotypes. However, in ZnO-NP treatments, high activity of APX was found in both Kargi (20.36–44.56%) and CSR 30 (26.32–47.37%) *O. sativa* genotypes,

compared with those in saline and nonstressed treatments (Figure 7C). In Kargi, the level of APX was slightly decreased in all of the treatments, compared with that of CSR 30.

### 2.8.4. GR (EC 1.8.1.7)

Salinity-exposed leaves showed a high activity of GR enzymes by 3.55–10.44% in Kargi and 5.85–11.88% in CSR 30, compared with those under nonstressed conditions (Figure 7D). The addition of ZnO-NPs upregulated the activity of GR in Kargi (5.66–11.85%) and CSR 30 (6.96–12.38%) leaves (Figure 7D). Although all treatments showed a maximum activity of GR enzyme, higher activity was demonstrated in ZnO-NP-treated treatments, compared with that observed under saline conditions. In addition, the GR activity was slightly higher in CSR 30 than in Kargi for all treatments.

### 2.9. Impact of ZnO-NPs on Stomata and Trichome Morphology

Scanning electron microscopy (SEM) observations of Kargi and CSR 30 leaves showed deformation in the stomata and trichome morphology under NaCl-treated conditions (Figure 8). The images obtained using SEM indicated that the reformation phenomena of stomata (Figure 8 A,B) and trichomes (Figure 8 C,D) were higher in ZnO-NP-treated leaves than in saline- and nonsaline-treated leaves. Additionally, more trichomes were recorded in ZnO-NP-treated Kargi and CSR 30 than stressed and nonstressed *O. sativa* leaves.

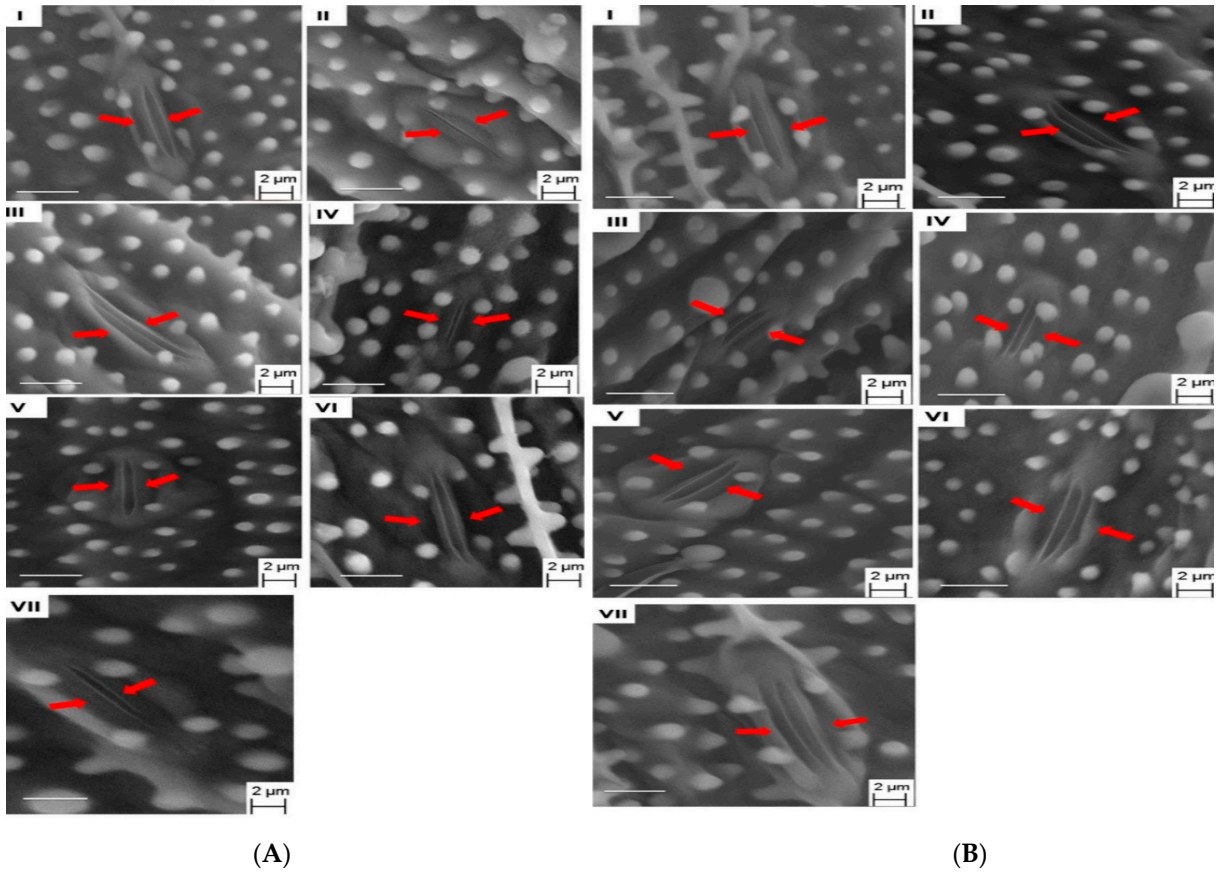

(**A**)　　　　　　　　　　　　　　　　　　　　(**B**)

**Figure 8.** *Cont.*

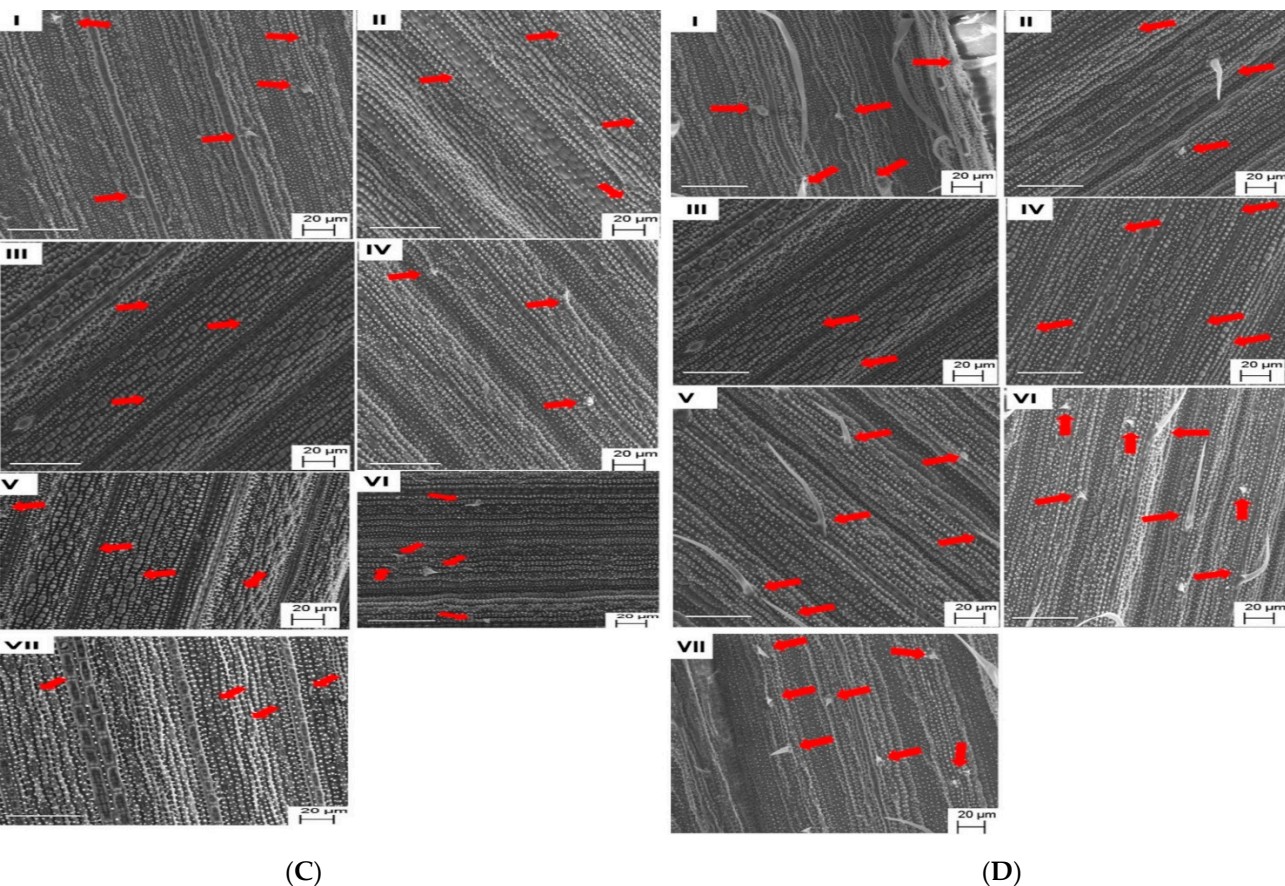

(**C**)                                            (**D**)

**Figure 8.** Impact of ZnO-NPs on stomata ((**A**) Kargi, (**B**) CSR 30) and trichome morphology ((**C**) Kargi, (**D**) CSR 30). Subfigures represent different treatments with ZnO-NPs: (**I**)—control, (**II**)—60 mM NaCl, (**III**)—80 mM NaCl, (**IV**)—100 mM NaCl, (**V**)—60 mM NaCl$^+$ ZnO-NPs (50 mg/L), (**VI**)—80 mM NaCl$^+$ZnO-NPs (50 mg/L), and (**VII**)—100 mM NaCl$^+$ ZnO-NPs (50 mg/L); red arrows indicate trichomes and stomata.

## 3. Discussion

In the past few years, the application of nanotechnology has rapidly increased in agriculture, including in abiotic stress tolerance approaches [35]. Applying ZnO-NPs significantly increased resistance against salinity stress in Kargi and CSR 30 genotypes of *O. sativa* by improving the biosynthesis o photosynthetic pigments, increasing the uptake of K$^+$ ions, and reducing the level of Na$^+$ ions, ROS, H$_2$O$_2$, and MDA by enhancing the activity of antioxidant enzymes [35,60]. The finding of Hafeez et al. [61] also suggested that Zn facilitates the regulation and stabilization of cell membranes by binding with phospholipids and sulfhydryl groups, mainly when plants experience stress conditions. Research findings in various crops revealed a positive impact of ZnO-NPs by improving physiological and biological actives under abiotic stress, e.g., heavy metal, salinity, chilling, and drought stress [35,62–68]. Furthermore, ZnO-NPs can contribute to forming organic compounds such as IAA and GA3, directly participating in plants' growth [69]. It also maintains the membrane structural stability, protein and enzyme activation, and cell prolongation and provides the tolerance mechanism against different types of ecological stress. These positive effects of ZnO-NPs may explain why zinc nanotreatment can improve *O. sativa* growth under salinity stress [69–71].

On the other hand, many studies showed that applying ZnO-NPs also create phytotoxicity on crops that inhibit their growth and development [47,48,72]. For example, Lopez-Moreno et al. [73] demonstrated that the application of 2000 mg/L ZnO-NPs on *Glycine max* led to genotoxicity that affected the integrity of the DNA. Further studies

have suggested that low and control applications of ZnO-NPs positively affect plants under salinity stress. Hezaveh et al. [74] recommended a value of 20 mg/L to alleviate the effect of salinity stress on the *Brassica napus* cultivar at the flowering stage. Recently, Mogazy et al. [75] demonstrated that 50 mg/L ZnO-NPs with NaCl could enhance the growth and antioxidant activity in *Vicia faba* under salinity stress. Similarly, in our study, a low concentration of ZnO-NPs (50 mg/L) was used and did not exert a negative impact on *O. sativa* genotypes.

In abiotic stress biology, several realistic models or in vitro experimental results showed that applying ZnO-NPs could increase the germination and yield of cereal crops, including *O. sativa*. Zhang et al. [63] showed enhancement in the *O. sativa* yield, ranging from 2.5% to 11.8%, by applying ZnO-NPs. The low concentration of ZnO-NPs in the hydroponic system showed better growth of *Hordeum vulgare* [76]. In another study, Upadhyaya et al. [65] reported that exposure to *O. sativa* seeds with ZnO-NP (5–50 mg/L) showed excellent potential for better germination of *O. sativa*. The foliar application of ZnO-NPs (5.0 g/L) significantly improved the growth and yield of *O. sativa* [77]. These findings concluded that applying ZnO-NPs positively impacted rice under normal conditions at low dose additions. Thus, our findings on the impact of ZnO-NPs under salinity stress could be an addition to managing damage during the early stage of *O. sativa* growth.

The present study revealed that the addition of ZnO-NPs led to the regulation of the salinity tolerance in Kargi and CSR 30 under controlled conditions [78,79]. The concentration of Zn content decreased in saline treatments in both cultivars. The supply of external Zn in the form of ZnO-NPs improved the level of Zn in considered *O. sativa* genotypes. Our results correlate with the findings of another study indicating that the input of ZnO-NPs increased Zn content by 13.5–39.4% in *O. sativa*, compared with conventional fertilization applications, and is safe for human consumption. [49]. Our results also showed that the increased Zn content stimulated the growth of Kargi and CSR 30 seedlings under salinity stress. At the same time, other studies showed that, without applying ZnO-NPs, salinity reduced *O. sativa* growth by reducing the uptake of Zn [65,80–82]. Our results agree with earlier findings that reported external application of ZnO-NPs improved *O. sativa* growth [65,82,83].

Kargi and CSR 30 growth were affected because salinity disrupts the uptake mechanism of potassium nutrients and stores excessive $Na^+$ ions in both *O. sativa* genotypes leaves, decreasing the level of $K^+$ and increasing the $Na^+/K^+$ ratio. In living cells, potassium helps maintain the turgidity and regulation of essential enzyme activity of the cell. At the same time, the shortage of $K^+$ ions and the high $Na^+/K^+$ ratio inhibits the growth and development of plants [84–87]. Therefore, applying ZnO-NPs promoted better adaptation toward salinity stress in Kargi and CSR 30 by lowering the Na+ concentration and $Na^+/K^+$ ratio in their leaves and improving the $K^+$ ion concentration. Improving the $Na^+/K^+$ balance is a key trait of salinity tolerance for plants to better adapt under salinity stress conditions [85]. Similar results on *Zea mays* and *Gossypium* spp. showed that the application of ZnO-NPs lowered the $Na^+$ and $Na^+/K^+$ ratio, which enhanced $K^+$ concentration, resulting in improvement of growth under salinity stress [88,89].

Proline is the most prominent organic solute that acts as an osmolyte and maintains the cytosolic pH and osmotic condition of cells during salinity stress [90]. The current study observed increased proline content in NaCl-treated Kargi and CSR 30. It was also suggested that the amount of proline increased in *Trigonella foenum-graecum* and *Gossypium* spp. under salinity stress [89]. In our study, adding ZnO-NPs (50 mg/L) decreased proline accumulation. In *Abelmoschus esculentus* and *Citrus* spp., reduced proline was noted under saline conditions by applying ZnO-NPs [53]. These results correspond with our findings on *O. sativa* seedlings.

The higher accumulation of $Na^+$ triggers ROS production, leading to membrane lipid peroxidation. The final lipid peroxidation product is MDA during oxidative stress, damaging the plant's membrane [91–94]. In this study, the higher MDA accumulation was found in Kargi and CSR 30 under saline conditions, and applying ZnO-NPs lowered

its accumulation under salinity conditions. The decrease in MDA content could be an adaptation process of *O. sativa* seedlings under salinity stress. It was explained earlier that applying ZnO-NPs can play a defensive role in coping with salinity stress by reducing the concentration of MDA, which helps in the improvement and permeability of the membrane and oxidative stress in plant seedlings [95]. Few findings also suggested that salinity stress disrupted the lipid and protein composition of the membrane that affected the architecture of the leaves' cell membrane [93,94,96–98].

Salinity directly affects the chloroplast structure and denatures their membrane and the rate of photosynthesis, which results in low grain formation and poor yield [56,99]. Furthermore, salinity stress alters the function of stomata by reducing their structure, process, and density [100]. This alteration reduces the rate of $CO_2$ uptake, decreases the photosynthesis rate, and the number of chloroplasts that damage the grana and thylakoid membrane directly hit the photosynthetic pigment–protein machinery of photosynthesis [101–104].

In current findings, the photosynthetic pigments—namely, Chl a, Chl b, total chlorophyll, and carotenoids—showed a remarkable reduction in Kargi and CSR 30 under salinity stress. At the same time, the application ZnO-NPs under saline conditions enhanced chlorophyll a and b, carotenoid, and total chlorophyll. Our finding was supported by those of Alabdallah et al. [53] and Venkatachalam et al. [105], which revealed that ZnO-NPs improved the photosynthetic pigments in *A. esculentus* and *Gossypium* spp. under saline conditions. This result of enhancement in photosynthetic pigment content in ZnO-NP-treated Kargi and CSR 30 may be due to the upregulation of ROS-scavenging enzymes that rapidly degrade the ROS radicals present in thylakoid membranes of chloroplasts [94,106] and likely another reason supporting current findings, i.e., reduction in the $Na^+/K^+$ ratio in both ZnO-NP-treated *O. sativa* genotypes helps to maintain ionic homeostasis of cells and upregulation of chlorophyllase activity [107,108].

An excessive accumulation of NaCl creates oxidative stress that produces ROS, e.g., $O_2^{\bullet-}$, $^1O_2$, $H_2O_2$, and $OH^\bullet$ [109–111]. ROS is destructive and can damage the cell and its organization. To overcome this problem, antioxidant enzymes (SOD, CAT, APX, and GR) come into play and help detoxify ROS radicals [22]. In this study, SOD, CAT, APX, and GR activity were higher in Kargi and CSR 30 under saline conditions. Furthermore, the higher activity of these antioxidant enzymes indicated the greater adaptation of Kargi and CSR 30 under salinity stress.

Similarly, rice under salinity stress was found to have increased antioxidant enzyme activity [96]. Further, in our study, the applications of ZnO-NPs upregulated the antioxidant enzyme activity more in ZnO-NP-treated plants than those grown in normal saline conditions without NPs. Similarities to our results were also found in results indicating that the application of ZnO-NPs upregulated the activity of antioxidant enzymes in *A. esculentus* and *Z. mays* under salinity stress [53]. Salinity affected the morphology of the stomatal aperture and trichome of plant leaves under salinity stress [112–116]. Reduction in the morphology of stomata and trichome affects the uptake of $CO_2$ from the atmosphere, reducing the photosynthesis rate [76]. An improvement in the morphology of trichomes and structure and density of stomata were noted in Kargi and CSR 30 under salinity with the addition of ZnO-NPs. The application ZnO-NPs improved the morphology of the stomatal aperture and trichome in spring barley under hydroponic conditions. However, the authors also showed that these subcellular organelles were damaged with high concentrations of NPs [76].

## 4. Materials and Methods

Commercial-grade zinc oxide (ZnO) nanopowder (particle size < 50 nm; batch no, 6063960) was purchased from Sisco Research Laboratories (SRL.) Pvt. Ltd., Maharashtra, India, which was used following a previously described protocol [77]. The seeds of Kargi landrace (IC-0642936) were collected from a farmer of Pauha village Machhlishahr, Jaunpur, Uttar Pradesh, India, and CSR 30 salt-tolerant basmati cultivar was obtained from Basmati Export Development Foundation, Meerut, India. The genotype CSR 30 is a salt-tolerant type

of basmati rice, immensely popular among farmers and released by Central Soil Salinity Research Institute (CSSRI), Haryana, India. The variety has the potential to grow under saline and sodic environments [117]. In contrast, the other genotype in this study, Kargi, is an undocumented, rare, and on-the-verge-of-extinction rice landrace mostly cultivated in saline lands in the Eastern part of Uttar Pradesh, India. Farmers who traditionally cultivated Kargi *O. sativa* claimed it can grow in saline areas. For this reason, documentation of this landrace becomes more important for crop improvement and rice production in saline areas.

Kargi and CSR 30 seeds were sterilized with 5% sodium hypochlorite solution for 30 min and washed with distilled water. After washing, 25 seeds of both rice genotypes were transferred to a round piece of wet filter paper in clean Petri dishes and kept in an incubator at 30 °C for germination for five days. All treatments were performed in triplicate. After 5 days, 10 rice seedlings of Kargi and CSR 30 were transferred into large-size Borosil culture test tubes (BRL_9820U1, O.D. × Length: 38 × 200) (in triplicate) containing 50 mL of half-strength of modified Hogland's solution with ZnO NPs (50 mg/L) (ZnO-NPs dissolved well into half-strength of modified Hogland's solution) and NaCl (60 mM, 80 mM, 100 mM) or without ZnO-NPs only have different concertation of NaCl (60 mM, 80 mM, 100 mM), test tubes were placed in a controlled growth chamber. The growth chamber was set at $28 \pm$ one °C with photon flux density 300–500 $\mu Em^{-2} s^{-1}$, relative humidity 75–80%, and 14 h/10 h photoperiod. After two weeks, plants were harvested, washed with double distilled water, and used for further analysis.

Details of the treatments are given below:

- Control (rice grown under nonstress conditions);
- Rice grown under saline conditions (60 mM NaCl Concentration);
- Rice grown under saline conditions (80 mM NaCl Concentration);
- Rice grown under saline conditions (100 mM NaCl Concentration);
- Application of ZnO NPs (50 mg/L) + salt stress (60 mM NaCl Concentration);
- Application of ZnO NPs (50 mg/L) + salt stress (80 mM NaCl Concentration);
- Application of ZnO NPs (50 mg/L) + salt stress (100 mM NaCl Concentration).

### 4.1. Growth Parameters

The treated *O. sativa* seedlings (in triplicate) were collected, and plant height (cm) of the different treatments was measured using a meter scale. Images regarding the growth of (A) Kargi and (B) CSR 30 rice genotypes after two weeks of treatments are presented in Supplementary File S1.

### 4.2. Estimation of Na$^+$ and K$^+$ Concentrations

The Na$^+$ and K$^+$ concentrations in the leaves of Kargi- and CSR-30-treated *O. sativa* genotypes were determined using a flame photometer (Elico-CL36, Hyderabad, India), as described by Abdelhamid et al. [79]. The dried rice leaves were placed in 1 N HCl for 12 h, and the concentrations of Na$^+$ and K$^+$ were estimated from the Na$^+$ and K$^+$ standard curves.

### 4.3. Estimation of Proline Content

The proline content of Kargi and CSR 30 fresh leaves was estimated per the protocol of Bates et al. 1973 [118]. Fresh Kargi and CSR 30 leaves (500 mg) were extracted in a sulfosalicylic acid solution, adding an equal amount of ninhydrin and glacial acetic acid solution. The sample was heated at 100 °C and added 5 mL of toluene, and the absorption was recorded at 520 nm using toluene as a blank. Proline is expressed as µg/g FW.

### 4.4. Estimation of Hydrogen Peroxide (H$_2$O$_2$)

The levels of H$_2$O$_2$ in Kargi and CSR 30 leaves were determined based on the study of Velikova et al. [119]. Leaves of Kargi and CSR 30 (0.5 g) genotypes of O. sativa seedlings were homogenized in an ice bath with 5 mL TCA (0.1% *w/v*). Centrifuged at 12,000× *g* for 15 min. After that, in 0.5 mL of the supernatant, 1 mL 1 M KI and 0.5 mL to 10 mM

potassium phosphate buffer with pH 7.0 were added. The absorbance of the supernatant was measured at 390 nm.

### 4.5. Measurement of MDA Content

MDA content in *O. sativa* leaves was measured using the thiobarbituric acid (TBA) method, as described by Assaha et al. [120]. Fresh Kargi and CSR 30 leaves (0.1 g) were homogenized with an extraction buffer (10 mM HEPES, pH 7.0, 15% tricarboxylic acid, 0.375% thiobarbituric acid, 0.25 N HCl, 0.04% butylated hydroxyl toluene, and 2% ethanol), incubated at 95 °C and then centrifuged. The supernatant was read between 532 nm and 600 nm. Further, the MDA content was calculated using the extinction coefficient $(155 \text{ mM}^{-1} \text{ cm}^{-1})$ [121].

### 4.6. Estimation of Zn Content

To analyze Zn content in treated *O. sativa* seedling leaves, they were ground into a fine powder and weighed; then, 0.5 g ground samples were transferred to 10 mL of a mixture of nitric acid and perchloric acid at a ratio of 2:1. This pre-digestion mixture was left overnight. After 24 h, these flasks were placed on a hot plate, and the sample was heated at 150–235 °C until the time at which orange fumes were converted into white fumes. The color change indicated complete digestion of the leave samples. Further, it was filtered by adding 2–3 mL of deionized water into 50 mL glass volumetric flasks and making up the volume by adding more deionized water into 100 mL flasks. The filtered extract was analyzed on an atomic absorption spectrometer (GBC-SavantAA-01-1006-03, Mumbai, India) [82].

### 4.7. Estimation of Photosynthetic Pigments

For observation of photosynthetic pigment contents (chlorophyll (Chl) a, Chl b, and carotenoids) in Kargi and CSR 30 *O. sativa* leaves, 100 mg of fresh leaves of both genotypes were weighed and ground into 10 mL of 80% (*v/v*) acetone. After grinding filter extract into 50 mL of the volumetric flask with the help of Whatman filter paper and funnel, the final volume was made with the help of 80% acetone [122,123]. Next, the absorbance of the extract was read at 645, 663, and 470 nm using acetone (80%) blank by using the following method:

$$\text{Chl-a (mg/g)} = 12.25 \, (\text{O. D}_{663.2}) - 2.79 \, (\text{O. D}_{646.8})$$

$$\text{Chl-b (mg/g)} = 21.50 \, (\text{O. D}_{646.8}) - 5.10 \, (\text{O. D}_{663.2})$$

$$\text{Total Chl (mg/g)} = 7.15 \, (\text{O.D}_{663.2}) + 18.71 \, (\text{O. D}_{646.8})$$

$$\text{Carotenoids (mg/g)} = [1000 \, (\text{O.D}_{470}) - 1.82 \, \text{Chl-a-}85.02$$

$$\text{Chl-b]/198}$$

where OD indicates optimal density.

### 4.8. Estimation of Antioxidant Enzymes' Activity

In Kargi and CSR 30 leaves, the SOD, CAT, and APX activities were measured according to the modified method of Takagi and Yamada [124]. One milliliter of crude enzyme extracts in a dialysis membrane was dialyzed in 500 mL of 10 mM K-P buffer (pH 7.8) for 12 h and renewed the K-P buffer every three hours. After that, this dialyzed extracted enzyme was used to measure the SOD activity at 560 nm absorbance.

The activity of CAT was measured using the previously described method by Tanaka et al. [125]. First, the CAT activity levels in Kargi and CSR 30 leaves were measured in 1 mL of assay mixture containing 100 mM K-P buffer (pH 7.8), 20 mM $H_2O_2$, and 2% (volume/volume (*v/v*)) crude enzyme extracts. After that, the activity of the CAT enzyme was measured by determining the amount of scavenged $H_2O_2$ per minute, and this was measured at specific absorbance of 230 nm.

The activity of APX was measured in 1 mL of assay mixture that had 100 mM K-P buffer (pH 7.8), 0.5 mM L-ascorbic acid, and 2% ($v/v$) crude enzyme extracts. Afterward, the oxidized L-ascorbic acid per minute was measured at a specific 290 nm absorbance.

The activity of GR was measured in 1 mL of assay mixture containing 50 mM K-P buffer (pH 7.8), 0.1 mM ethylenediaminetetraacetic acid (EDTA), 0.02 mM reduced nicotinamide adenine dinucleotide phosphate (NADPH), 0.02 mM oxidized form of glutathione, and 5% ($v/v$) crude enzyme extract. After that, NADPH's oxidized form was measured at a specific 340 nm absorbance to determine GR's activity.

### 4.9. Stomata and Trichome Morphology Observed via Scanning Electron Microscopy

The response and morphology of rice genotype CSR 30 and Kargi stomata and trichome under the different treatments of NaCl and ZnO-NPs were investigated using SEM (Zeiss EVO LS10, Oberkochen, Germany) in high vacuum water-free and covered with a thin silver metal layer at 20 kv.

### 4.10. Statistical Analyses

The statistical analyses were performed by one-way ANOVA using SPSS software 16.0 Version, and every experiment's technical specifications were triplicated. Significant differences in means were compared using Duncan's test at a significance level of $p \leq 0.01$.

## 5. Conclusions

This study highlighted that the salinity tolerance in *Oryza sativa* depends on $Na^+$ exclusion from leaves, balancing the $Na^+/K^+$ ratio, lowering the concentration of MDA, closing the stomata to avoid excessive water loss, and accelerating the activity of the antioxidant enzymes to cope-up stress. The application of ZnO-NPs upregulates salinity adaption processes. A comparative study between Kargi and CSR 30 under saline conditions demonstrated that Kargi rice genotypes were slightly less adapted under saline conditions because CSR 30 genotypes of *O. sativa* had better stomatal and trichome morphology and slightly higher antioxidant enzyme activities, photosynthesis pigments content, and Zn accumulation in all the treatments. However, with ZnO-NPs application, Kargi performed better under saline conditions. Cultivation of Kargi in saline areas with applying ZnO-NPs may result in better performance and production. Moreover, real-world field experiments on the performance of ZnO-NPs in the Kargi landrace, especially on degraded soils, should be performed in detail. Furthermore, in *O. sativa*, few genes are correlated with antioxidant enzyme activity, which works under environmental stress conditions. Therefore, expression analysis of genes that encode antioxidant enzymes related to the salinity tolerance of Kargi—namely, $Na^+$ and $K^+$ transporters—should be investigated.

## 6. Patents

National Identity or IC Number (Indigenous Collection), IC-0642936, of landrace rice genotype Kargi was allotted by the National Bureau of Plant Genetic Resources-ICAR New Delhi, India.

**Supplementary Materials:** The following supporting information can be downloaded at: https://www.mdpi.com/article/10.3390/agriculture12071014/s1.

**Author Contributions:** Conceptualization, A.S. and R.S.S.; methodology, A.S. and R.S.S.; software, A.S. and R.S.S.; validation, R.K.S. and V.D.R.; formal analysis, A.S. and R.S.S.; data curation, writing—original draft preparation, A.S., R.S.S., V.D.R. and T.M.; visualization, A.S. and R.S.S.; supervision, R.S.S. All authors have read and agreed to the published version of the manuscript.

**Funding:** This research received no external funding.

**Institutional Review Board Statement:** Not applicable.

**Informed Consent Statement:** Not applicable.

**Data Availability Statement:** The datasets used and analyzed during the current study are available from the corresponding author on reasonable request.

**Acknowledgments:** The authors thank the central facilities of the National Institute of Plant Genome Research, New Delhi, India, for providing SEM facilities.

**Conflicts of Interest:** The authors declare no conflict of interest.

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
