# Peer review of "Zinc Oxide Nanoparticles Improve Salt Tolerance in Rice Seedlings by Improving Physiological and Biochemical Indices"

_agriculture, doi:10.3390/agriculture12071014_

Round 1

Reviewer 1 Report

I found the manuscript entitled “Impact of ZnO Nanoparticles to Induce Salt Tolerance in Rice 2 Seedlings by Improving Physiological and Biochemical Indices”, quite interesting to read on rice sustainable production. The manuscript is good in structure; however, in my opinion, there are several issues (listed below) that need to be addressed to better emphasize the contribution of this study to the current knowledge.

1-      The authors should include more precise and recent literature in the manuscript and delete the old ones to increase the worth of this article, since there are 110 references.

2-      Add some numerical results in abstract.

3-      Keywords should be similar to title and abstract. Please change them according. Be specific.

4-      I suggest adding another treatment including ZnO Nanoparticles alone to make the differences clear.

5-      Add more details about the control treatment.

6-      Please also put some more details about the ZnO Nanoparticles.

7-      In the figures (1-7), what does the letter “a”, “b”, “c” and “d” indicate on the error bars.

8- Data values are well clear in table and figures.

9-  Discussion structure seems better but need to support your results in a comparative manner from recent literature.

10-  Check all references formatting.

Author Response

Reviewer-1

Response to the Reviewer’s comments

Author(s) are grateful to both the reviewer and editor for a careful and helpful analysis of our manuscript. We have corrected the MS in accordance with the comments and suggestions. Changes can be track in track change version file. Responses are added below.

Page number

Line number

Reviewer’s Comment

Author’s Revision

The authors should include more precise and recent literature in the manuscript and delete the old ones to increase the worth of this article, since there are 110 references.

References list is updated

1

25

Ad Add some numerical results in the abstract.

As per suggestion add some numerical results in the abstract. (Changes are highlighted with yellow mark)

1

37-38

The authors have rewritten the keywords similar to title and abstract. (Changes are highlighted with yellow mark)

14

What characteristic of ZnO NPs, is it directly dissolved in Hogland’s solution? Authors should mention it, because I do not understand how ZnO NPs affect the rice if it can’t be dissolved in Hogland’s solution.

ZnO NPs solution was prepared and ultrasonicated to avoid aggregation and immediately was added Hogland’s solution and mixed completely. 

We are thankful for your kind suggest. The experiment is completed and due to completion of plant authors are unable perform this addition treatment. We have justified the role of ZnO NPs on plants; especially on Rice and compared in NaCl stress condition.

Please also put some more details about the ZnO Nanoparticles.

More detailed are added.

 3-10

123-268

In the figures (1-7), what does the letter “a”, “b”, “c” and “d” indicate on the error bars.

Different small alphabets indicate a statistically significant difference between treatments with respect to control

(Changes are highlighted with yellow mark)

Data values are well clear in table and figures

Authors are highly thankful to reviewer for their appreciation

12

Discussion structure seems better but need to support your results in a comparative manner from recent literature.

Authors are highly thankful to reviewer for their appreciation. We have add recent literature.

(Changes are highlighted with yellow mark)

17-26

Da. Check all references formatting.

Authors are formatted and checked the all references

(Changes are highlighted with yellow mark)

Reviewer 2 Report

Reviewed manuscript “Impact of Zno nanoparticles to induce salt tolerance in rice seedlings by improving physiological and biochemical indices is an original and interesting study. Authors comprehensively evaluated the ZnO NPs effect on Rice, hence the study is beneficial for agriculture. I would suggest minor revision.

Following are some suggestions for further improvements:

I would suggest to alter the title as it seems complex. A suggestion would be: Zinc oxide Nanoparticles improve Salt Tolerance in Rice Seedlings by Improving Physiological and Biochemical Indices

Keywords should be full words.

In some figures units, avoid using “/” e.g mg/g should be mg g-1

Line 271: Remove commas from the line

Introduction and Discussion section needs to further strengthen by latest studies on the subject.  https://doi.org/10.1016/j.plaphy.2021.01.028 https://doi.org/10.1016/j.jhazmat.2021.127891

Proline and antioxidants role should be discussed in detail in the discussion and introduction https://link.springer.com/article/10.1007/s00425-019-03293-1

At some places in the text, there are grammatical mistakes that needs to be corrected by some native English colleague.

To further strengthen introduction, following latest studies etc. are suggested to cite:

https://dx.doi.org/10.3389%2Ffpls.2021.770084

https://doi.org/10.1016/j.plantsci.2019.110270

https://doi.org/10.1080/11263504.2020.1762783

Author Response

Reviewer-II

Response to the Reviewer’s comments

All the reviewer`s comments are reasonable, and we have corrected the MS in accordance with the comments and suggestions. We believe current version will get acceptance for publication. We are thankful for reviewers’ positive remarks

Page number

Line number

Reviewer’s Comment

Author’s Revision

Reviewed manuscript “Impact of Zno nanoparticles to induce salt tolerance in rice seedlings by improving physiological and biochemical indices is an original and interesting study. Authors comprehensively evaluated the ZnO NPs effect on Rice, hence the study is beneficial for agriculture

Authors are highly thankful to the reviewer for their kind words and comments.

1

1-2

I would suggest altering the title as it seems complex. A suggestion would be: Zinc oxide Nanoparticles improve Salt Tolerance in Rice Seedlings by Improving Physiological and Biochemical Indices

As per your valuable suggestion authors are change the title of the article

1

37-38

Keywords should be full words.

For better understanding authors are write full words of keywords (Change highlight with yellow mark)

4-9

165-264

In some figures units, avoid using “/” e.g mg/g should be mg g-1

As per your suggestion, I make changes to every figure

11-12

313-317

Line 271: Remove commas from the line

As per your valuable suggestion remove the commas

17

602, 651

Introduction and Discussion section needs to further strengthen by latest studies on the subject.  https://doi.org/10.1016/j.plaphy.2021.01.028 https://doi.org/10.1016/j.jhazmat.2021.127891

Add suggested paper as reference number 2 , 20 in the introduction and discussion

(Change highlight with yellow mark)

2

74-76

Proline and antioxidants role should be discussed in detail in the discussion and introduction https://link.springer.com/article/10.1007/s00425-019-03293-1

Add suggested paper as reference number 19 in the introduction and discussion

(Change highlight with yellow mark)

At some places in the text, there are grammatical mistakes that needs to be corrected by some native English colleague.

Authors are rewrite all the error related to english

1, 2,  17, 20

41-46, 63-65, 96, 609-611, 706-707 , 630-632

To further strengthen introduction, following latest studies etc. are suggested to cite:

https://dx.doi.org/10.3389%2Ffpls.2021.770084

https://doi.org/10.1016/j.plantsci.2019.110270

https://doi.org/10.1080/11263504.2020.1762783

Add suggested paper as reference number 5, 39, 12  in the introduction section

Reviewer 3 Report

The research work is conducted on a sound scientific basis, however, I wonder if the authors have discussed any novelty of this study. Zinc Oxide Nanoparticles (ZnO-NPs) are known to alleviate various abiotic stress, including salinity stress in different crops. Several papers are already published to support this conclusion. Also, the authors should include the toxicity data, if any, while using the nanoparticles in their study.

Author Response

Reviewer-III

Round-I

Author(s) are grateful to both editor and reviewer for a careful and helpful comments on our manuscript.

Response to the Reviewer’s comments

Page number

Line number

Reviewer’s Comment

Author’s Revision

The research work is conducted on a sound scientific basis, however, I wonder if the authors have discussed any novelty of this study.

Thank you for your positive comments; we have clearly marked novelty in current, and we are sorry it was not clear in previous version

Zinc Oxide Nanoparticles (ZnO-NPs) are known to alleviate various abiotic stress, including salinity stress in different crops. Several papers are already published to support this conclusion. Also, the authors should include the toxicity data, if any, while using the nanoparticles in their study.

Thank you so much, we have discussed possible literature results and compared with our findings. We have added low dose of NPs under stress condition to evaluate adaptation mechanism of plants. Toxic effects are discussed based on earlier finds, this low concentration added in our experiment did not show toxic effects.

Reviewer 4 Report

The present study highlighted the positive regulatory role of ZnO-nanoparticles in imparting salinity tolerance in rice. The major mechanisms included: Na+ exclusion from rice leaves, balancing the Na+/K+ ratio, lowering the concentration of MDA, stomata closure to avoid the excessive water loss and enhanced activity of the antioxidant enzymes. The study is valuable and findings are significant; however, I have some queries and suggestions

1.      Why both cultivars of salt-tolerant nature were included in this experiment? It would have been much better to include contrasting (salt sensitive and salt tolerant) cultivars for better mechanistic insights on ZnO-induced tolerance.

2.      How the concentration of ZnO NPs (50 mg/L) was standardized and why it has not been mentioned?

3.      Why ZnO-NP alone treatment as control is not included in this experiment.

4.      The seeds were sterilized with 0.01% HgCl2 for 2 minutes. Is it the appropriate concentration/chemical for surface sterilization?

5.      Section 2.1 require may be supplemented with good quality comparative photographs.

Minor comments

1.      The first introductory sentence in line 40 is incomplete. Please correct grammatical errors in lines 40-43

2.      What are resulting effects on membrane integrity? Line 51

3.      Line 56: glycophytes

4.      Incomplete sentence in lines 68-69

5.      Line 91: the present was aimed? Add missing part.

6.      Line 99-101: Seems that ZnO was responsible for height reduction. Please rephrase.

7.      Line 131: less reproduction of Na? change it.

8.      Line 137-140 rewrite these findings for better clarity.

9.      Line 418: described previously protocol? Modify.

10.  Line 250-252: mentioning about APX? Kindly recheck.

11.  Fig 8 caption elaborate and correct the typographical error.

12.  Delete lines 288-289

13.  Rephrase line 454-456.

14.  Carefully remove the typographical error from the discussion section.

Author Response

Reviewer-IV

Dear Editor

Reviewer`s comments, has been significantly improved our manuscript. Authors are thankful for kind support. Responses are added below and file suppled with track change mode

Response to the Reviewer’s comments

Page number

Line number

Reviewer’s Comment

Author’s Revision

The present study highlighted the positive regulatory role of ZnO-nanoparticles in imparting salinity tolerance in rice. The major mechanisms included: Na+ exclusion from rice leaves, balancing the Na+/K+ ratio, lowering the concentration of MDA, stomata closure to avoid the excessive water loss and enhanced activity of the antioxidant enzymes. The study is valuable and findings are significant

The authors are thankful for anonymous for critical positive evaluation. The authors edited thoroughly whole manuscript for clearly understandable the results and novelty of present work. All the typos and grammatical errors are carefully checked and corrected. Suggested papers are carefully evaluated and discussed with present findings.

Why both cultivars of salt-tolerant nature were included in this experiment? It would have been much better to include contrasting (salt sensitive and salt tolerant) cultivars for better mechanistic insights on ZnO-induced tolerance.

ZnO NPs are well explored to support crops including rice. Our treatments are performed and compared with two rice variants i.e., Kargi (Undocumented, locally grown and claimed salt tolerant, another CSR 30 which well popular salt tolerant variety in north India). The authors believed that the current version will meet your expectation and get approval for further processing.    

12

344-346

How the concentration of ZnO NPs (50 mg/L) was standardized and why it has not been mentioned?

Already many more articles explained that  ZnO NPs (50 mg/L) was standardized and we add article of  Mogazy et al. [75] demonstrated , 50 mg/L ZnO-NPs with NaCl can enhance the growth and antioxidant activity in Vicia faba  under salinity stress.

Why ZnO-NP alone treatment as control is not included in this experiment.

The program was already completed that's why cannot add ZnO-NP control b ut many more articles were already mentioned that show positive effects ZnO-NPs as control under salinity stress.

14

454

The seeds were sterilized with 0.01% HgCl2 for 2 minutes. Is it the appropriate concentration/chemical for surface sterilization?

Corrected,  5% sodium hypochlorite solution for 30 min

(Highlighted section with yellow colour)

Section 2.1 require may be supplemented with good quality comparative photographs

Add the clear photograph

1

41-46

The first introductory sentence in line 40 is incomplete. Please correct grammatical errors in lines 40-43

This section is rewritten

(Highlighted section with yellow colour)

2

58-61

What are the resulting effects on membrane integrity? Line 51

This section is rewritten

(Highlighted section with yellow colour)

2

68

Line 56: glycophytes

Corrected

(Highlighted section with yellow colour)

2-3

85-103

Incomplete sentence in lines 68-69

This section is rewritten.

(Highlighted section with yellow colour)

3

107

Line 91: the present was aimed? Add missing part.

Corrected

(Highlighted section with yellow colour)

3

116

Line 99-101: Seems that ZnO was responsible for height reduction. Please rephrase.

Corrected

(Highlighted section with yellow colour)

5

153-155

Line 131: less reproduction of Na? change it.

This section is rewritten.

(Highlighted section with yellow colour)

5

163-165

Line 137-140 rewrite these findings for better clarity.

This section is rewritten.

(Highlighted section with yellow colour)

15

481

Line 418: described previously protocol? Modify.

Corrected

(Highlighted section with yellow colour)

10

297

Line 250-252: mentioning about APX? Kindly recheck.

Corrected

(Highlighted section with yellow colour)

11-12

314-317

Fig 8 caption elaborate and correct the typographical error.

Corrected

(Highlighted section with yellow colour)

11

350

Delete lines 288-289

Rewrite with appropriate meaning

15-16

517-521

Rephrase line 454-456.

Rephrase

(Highlighted section with yellow colour)

Carefully remove the typographical error from the discussion section.

Carefully check all the typographical error from the discussion section.

Round 2

Reviewer 3 Report

The authors have improved their writing of the manuscript and have added information about NP toxicity, hence I recommend the article for publication.

Reviewer 4 Report

The manuscript has been substantially improved on the suggested lines. Thank you for the necessary corrections and proper justification of the queries.